# The Aerodynamic Effect of an Alula-like Vortex Generator on a Revolving Wing

**DOI:** 10.3390/biomimetics7030128

**Published:** 2022-09-10

**Authors:** Ping-Han Chung, Po-Hsiang Chang, Szu-I Yeh

**Affiliations:** Department of Aeronautics and Astronautics Engineering, National Cheng Kung University, Tainan 701, Taiwan

**Keywords:** alula, vortex generator, leading-edge vortex, PIV, revolving wing

## Abstract

An alula is a small structure of feathers that prevents birds from stalling. In this study, the aerodynamic effect of an alula-like vortex generator (alula-VG) on a revolving wing was investigated using the PIV technique in a water tank. The alula-VG was mounted on a rectangular wing model at two spanwise positions. The wing model with a revolving motion was installed at different angles of attack, which included pre-stall and post-stall conditions. The velocity fields around the wing model with/without an alula-VG were measured and analyzed, including the vorticity contour, the circulation of vortex structures, and the corresponding sectional lift coefficient, which are used to explain the aerodynamic effect induced by an alula-VG. The lift-off and bursting of the leading-edge vortex (LEV) affect the magnitude of the chordwise circulation and the section lift coefficient. The results show that compared to an alula-VG mounted fixed wing model, the flow interactions among the alula-VG induced spanwise flow, the inertial force caused by the revolving motion, and the wing-tip vortex play important roles in the vortex bursting and the resultant aerodynamic performance. The effect of an alula-VG on a revolving wing depends on its spanwise position and the angle of attack of a wing model, which need to be properly matched.

## 1. Introduction

Natural flyers have wings with a low aspect ratio (*AR*) and achieve excellent flight performance in a regime with a low Reynolds number (*Re*) [1,2]. Generating the leading-edge vortex (LEV) is the main and important aerodynamic mechanism that prevents flow separation at a high angle of attack for natural flyers. A strong LEV formed by a flapping motion produces a low-pressure region over the suction side—generally the upper surface—of the wing for the generation of lift [3]. A stably attached LEV inboard of a wing is achieved if there is a balance between the production of vorticity at the leading edge and the spanwise vorticity transport (the effects of the local Rossby number, *Ro*, and the Coriolis force) [2]. The generation and evolution of LEVs during the flapping motion is one of the most popular and promising research topics [4,5,6].

Quasi-steady revolving or rotation wing motions have been used to simplify and mimic the flapping motion in many studies, which is similar to a flapping wing in midstroke when it is sufficiently removed from the effects of either pronation or supination [7,8,9,10]. Manar et al. conducted experiments on rotating and translating wings at a high angle of incidence [11]. The strength of forces and vortices generated by the wings with rectilinear and rotating kinematics are similar. A steadily rotating wing at a constant angle of attack represents a simplified model of the propulsion system for a flapping wing. Bhat et al. investigated the LEV formation and its stability with different *AR*, *Re*, and *Ro* values [7]. Rotating wings of different *AR*s have a similar LEV structure at a constant *Re* with a characteristic length of span (*Re_b_*). There is an increase in the circulation of the LEV when the spanwise distance from the axis of rotation is increased. At a higher *Re*, the LEV is maintained over a larger region on the wing. An increase in the *Ro* results in a weaker LEV and shows weaker spanwise flow, which is because the Coriolis accelerations due to wing rotation are inversely proportional to the *Ro*.

The generation and evolution of LEVs depend on the wing geometry, *AR*, *Re*, and *Ro*. For natural flyers, the alula is the most common configuration used to stabilize the LEV during the flapping motion. An alula is a small structure of feathers that prevents birds from stall and flow separation [12,13]. It is similar to an outboard canted leading-edge flap, which induces an attached LEV on an unswept wing in steady translation [14]. The alula generates a small streamwise vortex, which energizes the boundary layer and lets the flow become more attached to the surface of the wing, even near the wingtip. The alula helps to generate lift even at a low *Re*. The streamwise location and the deflection angle of the alula are all important for flow control on the wing surface, which includes vortex steering and the evolution of the spanwise flow [15]. Heavier and larger birds have longer alulae to sufficiently suppress flow separation at high angles of attack [14]. In fact, the functions of alulae are similar to vortex generators, which are usually installed with an incidence angle to the local flow on the wing surface and act as a passive flow control device [16,17]. Vortex generators (VGs) induce streamwise trailing vortices that penetrate near-wall flows and delay boundary layer separation. By using VGs, there is a significant increase in lift and a reduction in drag for different airfoils and wing geometries. The factors that dominate the performance of VGs are their shape (vane type, wheeler type, wishbone type, etc.) [18] and height [19].

Previous studies on the aerodynamic effects of alulae or VGs have focused on the fixed wing flight condition. However, the function and the aerodynamic effect of alulae in flapping or revolving motion still lack further investigation. A revolving wing can demonstrate the instantaneous forces on a flapping wing during the translation motion. The alula–wing interaction in revolving motion is a worthy study topic in flapping/bionic aerodynamics that has not yet been studied. Generally, a flapping motion induces slight boundary layer separation compared to fixed-wing motion, which also leads to a higher stall angle of the flapping wing model. Beals and Johns determined that the stall angle for a revolving wing is approximately 35° [20]. In this study, the aerodynamic effect of an alula-like vortex generator on a revolving wing at different angles of attack (10°, 25°, and 45°) was investigated, including the regular flight, pre-stall, and post-stall conditions. The designed alula-VG was placed at the leading edge of a biomimic revolving wing, which mimics the geometry of natural flyers and the motion of flapping wings in midstroke. According to the dimensional analysis of birds, the particle image velocimetry (PIV) flow visualization experiments were conducted in a water tank. The sectional velocity fields at different locations along the wingspan were measured for a detailed understanding of the overall flow structure. In addition, a corresponding quantitative flow analysis, which included the circulation around the LEV and the lift coefficient variation, was performed in this study. The results show that the flow interactions among the alula-VG induced spanwise flow, the inertial force caused by the revolving motion, and the wing-tip vortex play important roles in the vortex bursting and the resultant aerodynamic performance. The effect of an alula-VG depends on its spanwise location and the angle of attack of the wing model, which are discussed in detail in this manuscript.

## 2. Experimental Setup

### 2.1. Test Model Design

The research shows that the spanwise region with a stably attached LEV on a revolving wing is always less than 4 times that of the chord length at a 45° angle of attack and a wide *AR* range [21]. Therefore, a flat plate with *AR* = 4 at a size of 40 mm (chord length, *c*) × 160 mm (span, *b*) × 2 mm (thickness, *t*) was designed as the base model in this study. The base wing model had a rectangular planform with a value of *AR* = 4, and the ratio between the plate thickness and the wing chord length, *t/c*, was 0.05. The wing model was made of T6061 aluminum alloy and mounted on a rotary-driven shaft. The driven shaft was fixed by a roller bearing and driven by a stepper motor (Nema34, 86HSE12N-BC38, OMC, Nanjing, China). The stepper motor was controlled using an HSS86 closed-loop driver (Spark Motors, Maharashtra, India). The motor controller (Arduino Mega 2560) used micro-steps at a resolution of 3200 steps/rev. The root of the wing model was offset from the rotation axis (*b_0_*) by 0.12*b*. The value of *b_0_* should not exceed 0.5*b*, which shows a decrease in lift production in the previous study [7].

Linehan and Mohseni showed that the alulae for landbirds are positioned between 0.25*b* and 0.5*b* from the wing root, or *L_w_/b (L_w_*)* = 0.25, 0.5 [13]. Wang and Ghaemi determined that a vane-type VG had the greatest performance in stabilizing flow [22]. A previous study also showed that a ratio of 0.11 between the height of the alula and the chord length (*h** = *h/c*) produced the greatest increase in lift [19]. In consideration of the above-mentioned, the alula-VG designed for this study was 4.6 mm in chord length, 14.0 mm in span, and 0.5 mm in height. In addition, according to the statistical results of the birds [13], the *AR* of an alula-VG was designed as 3.04 and the area ratio of an alula-VG (VG_area_/Wing_area_) was 1%. The right front edge of the alula-VG was positioned at the leading edge of the wing and the incidence angle, *θ*, was 27°, which also fits the average value of natural flyers. The alula-VG was made of polylactic acid (PLA) and was 3D-printed by a DUAL-300 FDM3D printer (Ping, Hsinchu, Taiwan).

The value of *Re* is defined as three-quarters of the span, as follows:(1)$$Re_{b}=\frac{3\rho \omega b^{2}}{4\mu }$$
where *μ* is the dynamic viscosity of the water, *ρ* is the density of the water, and *ω* is the angular velocity (=3.14 rad/s) of a revolving wing. *Ro* is the ratio of the inertial force to the Coriolis force. Lentink and Dickinson determined that the Coriolis force promotes spanwise flow [9]. Bhat et al. showed that an increase in the *Ro* weakens the LEV and noted that *Ro_b_* (=*R_g_/b*) allows the flow structure and resultant aerodynamics to be characterized, where *R_g_* is the radius of the gyration of the wing [7]. The wing models with/without an alula-VG used in this study were all under the same revolving motion, with *Re_b_* = 6.8 × 10^4^ and *Ro_b_* = 0.55. The *Re_b_* was similar to that for birds, and the Coriolis force (rotating effect) was dominant at a low *Ro*. The schematic diagram of the wing model with an alula-VG and the definitions of the geometric parameters are all shown in Figure 1.

### 2.2. Particle Image Velocimetry (PIV) System

The flow field measurements were all conducted in an 1800 mm × 800 mm × 900 mm glass-walled water tank. The wing models, mounted on the rotary shaft at different angles of attack (10°, 25° pre-stall and 45° post-stall), were placed at the mid-depth of the tank filled with water. The minimum tip clearance with respect to the tank wall was 3.54*c* to minimize wall interference [23]. The azimuthal position of the wing followed a smoothed linear ramp [24].

PIV measurements were conducted to determine the flow behavior on the upper surface of the revolving wing. The whole experimental setup is shown in Figure 2. A high-speed camera (Photron Fastcam SA-X; 500 Hz; 1024 × 1024 pixels; 12 bits) was used to capture the continuous flow motion at a frame rate of 250 or 500 fps, depending on the plane location. The camera was equipped with a 60-mm Nikon lens (AF-S DX NIKKOR 18–200 mm f/3.5–5.6 G ED VR II). The flow measurement region was illuminated by a laser sheet, which was produced by a 5-W continuous laser (Elforlight HPG-5000; CW; 532 nm) with a set of optical lenses. The thickness of the laser sheet was about 1.5 mm [25]. The seeding particles used in the water were hollow glass spheres with an 8 to 12 μm diameter, from TSI Inc. The density of the seeding particles was around 1.10 ± 0.05 g/cm^3^ and the corresponding Stokes number was 3.13 × 10^−5^, which shows good response characteristics with water.

An alula-VG was positioned at 0.25*b* or 0.5*b* from the wing root. The paralleled x-z plane, with multiple different stations on the y-axis, was captured by the high-speed camera. The parameter *r** (*y/b*) represents the normalized y-position relative to the wingspan. The value of *r** from the wing root to the wing tip varied from 0 to 1. The parameter *r** was also used to define the local *Ro*. The equidistant measurement (laser) planes were located at *r** = 0.36–0.56 and 0.61–0.81 for the alula-VG positioned at 0.25*b* and 0.5*b*, respectively, which means the regions near the alula-VG were measured. The measurement planes were all in 0.05*b* increments, and there were 5 planes measured for a specific wing model. The schematic of the flow visualization planes and the spacing between each plane are illustrated in Figure 2. To minimize the amount of laser reflection, the wings were all coated with matte black paint.

An open-source code, PIVlab [26], was employed for calculating the flow field through the captured image pairs. It uses a multi-pass, multi-grid window deformation technique to calculate the velocity field of a fluid. The interrogation windows were 128 × 128 pixels (initial pass), 64 × 64 pixels, and 32 × 32 pixels (final pass), with an overlap of 50%. The velocity vectors of each interrogation window were evaluated using a cross-correlation method. Small sub-images of an image pair were cross-correlated to determine the most probable particle displacement in the interrogation area of interest. The LEV was identified using the *Q*-criterion [27]. The circulation inside this region (single LEV), *Γ_LEV_*, was calculated by integrating the vorticity. For sectional circulation, Jones et al. proposed a simple integration bound that is defined by a rectangular box (one chord length long and half a chord length high) that is directly attached to the suction side of a wing [28]. The area below the wing was omitted. The vorticity was integrated to determine the vortex strength or circulation, *Γ*. For a quantitative comparison, the normalized circulation on a chordwise plane, *Γ**, was calculated as *Γ/V_tip_c*, where *V_tip_* is the wing-tip velocity. The sectional lift coefficient, *C_l_* = 2*Γ/Vc*, was calculated via the Kutta–Joukowski theorem, where *V* is the local wing velocity.

## 3. Results and Discussion

### 3.1. The Base Case (without Alula-VG)

The flow field induced by a revolving wing helps to explain the corresponding flow structure, especially the LEV, at different x-z planes. The strength of the LEV is highly related to the strength of lift; the LEV structures shown at different spanwise planes also illustrate the lift distribution in the spanwise direction. As a control group, we discuss the flow field around the base wing model without an alula-VG; the instantaneous snapshots of the vorticity contour at *α* = 10° on different spanwise planes are shown in Figure 3. The LEV was stably attached to the region near the wing root (Figure 3a–c) and slightly lifted but still attached to the wing surface at other spanwise planes. The rise of the LEV in the spanwise direction indicates a tendency for flow separation at the outboard of the wing, but it is not obvious yet at a low angle of attack. The vorticity contours for the base wing model at *α* = 25° (pre-stall) on different spanwise planes are shown in Figure 4. The LEV was still attached to the wing surface in each plane and the size of the LEV slightly grew in the outer planes. The stably attached LEV was due to the strong rotating effect, which corresponds to the local *Ro*. The angular velocity was higher in the outboard plane than that in the inner plane. An LEV near the leading edge and a trailing-edge vortex (TEV) with a reversed spinning direction are clearly shown in the x–z plane on the outer wing sections (Figure 4c–f). Because stalling had not been shown in this pre-stall condition yet, the LEVs were stably attached to the wing surface on each plane. The results agree with those of the study by Jardin and Colonius [2], which showed that the LEV extended from the wing root to approximately three chords in the spanwise direction (*r** = 0.12–0.87 for this study). However, there was also a spanwise variation in the strength of the LEV. This is because root-to-tip flow (or a spanwise pressure gradient) appeared [29]. Through the vorticity contour on different x-z planes, the intensity of the vortex increased toward the wingtip. The peak vorticity of the LEV was located farther downstream and at the outboard of the wing.

At *α* = 45° (post-stall), the instantaneous snapshots of the vorticity contour on different spanwise planes are shown in Figure 5. The LEV formed and was attached to the wing surface only on the planes near the wing root. Positive vorticity was fed by the shear layer that emanated from the leading edge [28]. The LEV was extended in the chordwise direction (growth) but was limited to near the mid-chord of the wing at *r** = 0.36 and 0.46 (Figure 5a,b). The LEV lifted off in the outboard region of the wing. The combination of the larger local *Ro* and higher *α* had an adverse effect on the attachment of the LEV. Even for the revolving wing in stall, the LEV near the wing root region was stronger than that at *α* = 25°, which is consistent with the previous study [30]. This also shows that the suction force, which is the negative pressure on the upper surface of the wing, was greater under the area that was covered by the LEV. The LEV propagated farther downstream in the outer spanwise planes (*r** = 0.36–0.66). A bursting of the LEV was observed at *r** = 0.76 and 0.81 (Figure 5e,f), which dramatically destroyed the structure of the LEV. Lambourne and Bryer showed that the bursting of a vortex in a plane wing is related to the adverse pressure gradient that is associated with the existence of a trailing edge [31]. This occurred because the LEV was stabilized by the spanwise vorticity transport, which is the relative amplitude between the influx vorticity from the leading-edge shear layer and the coherent vortex. The location of the bursting LEV was associated with the stagnation point on the wing surface that was strongly related to the lift generated.

The values of *Γ* and *Γ** were calculated for the simplest integration bound (a rectangle box of 1*c* × 0.5*c*) that was directly attached to the suction side of the wing [28]. The total circulation of the LEV, *Γ_LEV_*, was calculated by integrating the vorticity contour around the boundary of the LEV which was identified by Q-criteria. For the base wing model, the variations of *Γ**** on different x-z planes at different angles of attack are shown in Figure 6. There was a slight increase in the value of *Γ**** in the outboard region at *α* = 10° and 25°. Both angles of attack were below the stall angle of the revolving wing (*α* = 35°) [21]. For the post-stall condition (*α* = 45°), a significant increase in the value of *Γ** happened, which was related to the growth of the LEV at a high angle of attack. However, in the outboard region of the wing, technically *r** = 0.71, there was a dramatic decrease in the *Γ** at both *α* = 25° and 45°. This is because there was a lower spanwise vorticity flux (the effect of the Coriolis force) and a zero-pressure gradient near the mid-span region, which induced the bursting of the LEV [7]. A lower value of *Γ** means less lift production near the mid-span region.

### 3.2. Flow Field of the Case with an Alula-VG

According to previous studies on fixed wings with an alula-VG, the alula produces a streamwise vortex, which helps to maintain and stabilize the LEV even over the outer region of the wing’s surface [15]. In this study, an alula-like VG was placed at *L_w_** = 0.25 and 0.50 on a revolving wing model to determine their aerodynamic effects on the development of the flow/vortex structure and the corresponding lift distribution, respectively.

The instantaneous snapshots of the vorticity contour on different spanwise planes for the wing model on which an alula-VG was placed at *L_w_** = 0.25 (*r** = 0.37) are shown in Figure 7 and Figure 8. The red mark represents the presence of an alula-VG in the measurement plane. In the pre-stall condition (*α* = 25°), there was a slight lift-off of the LEV, and the intensity of the LEV was larger on the x-z plane near the alula-VG (*r** = 0.36, Figure 7a) than in the case without an alula-VG (Figure 4a). A trailing-edge vortex (TEV) with a reversed spinning direction was visible in the region near the wing root, which is not shown for the base wing model. The increment in the strength of the LEV with the presence of an alula-VG was due to the induced streamwise and spanwise flux, and this resulted in an increase in the lift. The LEV burst quickly in the spanwise direction. This indicates that the presence of the alula-VG resulted in the downstream movement of the LEV. It is also noted that there is only a small vortex structure shown in Figure 7b,c which is the x-z plane just a little bit away from the region covered with an alula-VG. At *α* = 45° (post-stall), the presence of an alula-VG also resulted in a lift-off and stronger LEV at *r** = 0.36 (Figure 8a). Greater strength in the LEV indicates that, even in the post-stall condition, the presence of the alula-VG energized the LEV. At the outer planes (*r** = 0.46, 0.56), although there was vortex bursting, the vortices did not lift off. In addition, the LEV was attached to the wing surface even at the region of the outboard of the alula-VG (*r** = 0.56, shown in Figure 8c). The spanwise vorticity flux that was induced by the trailing vortex of the alula-VG resulted in a less concentrated LEV.

The instantaneous snapshots of the vorticity contour on different spanwise planes (*r** = 0.61–0.81) for the wing model on which an alula-VG was placed at *L_w_** = 0.50 (*r** = 0.62) are shown in Figure 9 and Figure 10. In the pre-stall condition (*α* = 25°), the presence of an alula-VG resulted in a larger LEV at *r** = 0.61, which was directly covered by the alula-VG. The LEV was also stably attached to the surface of the wing-tip region even under a vortex bursting halfway through the spanwise direction. However, in the post-stall condition (*α =* 45°), the LEV was very strong but slightly lifted off in the region near the alula-VG. The LEV was already broken into smaller structures and separated from the wing surface in the near wing-tip region (*r* =* 0.81, shown in Figure 10c). An alula-VG placed on the relatively outer position induced a strong flow interaction, which included the alula-VG-induced spanwise flow, the outboard velocity due to the rotational effect, and the wing-tip vortex. The flow interactions destroyed the LEV structure on the surface of the outboard region, which resulted in a reduction in the aerodynamic performance of the wing model.

For the wing model with an alula-VG at *L_w_** = 0.25, the spanwise distributions of *Γ** and *Γ_LEV_** are shown in Figure 11a. The value of the base wing model is shown in a solid symbol as the control group. At *α* = 10°, the usage of an alula-VG was not beneficial for the intensity of the vortex for both *Γ** and *Γ_LEV_** at every x-z plane. This is because the vortex induced by the alula-VG affected the spanwise vorticity flux and the spanwise pressure gradient, which weakened the strength of the LEV and reduced the value of circulation. At higher angles of attack (*α* = 25° and 45°), there were increases in the values of *Γ_LEV_** and *Γ** in the region near the alula-VG. Toward the outboard region of the wing, the bursting of the LEV and the higher local *Ro* due to the revolving motion gradually diminished the effect of the alula-VG.

The spanwise distributions of *Γ** and *Γ_LEV_** for the wing model with an alula-VG at *L_w_** = 0.50 are shown in Figure 11b. At α = 10°, the usage of an alula-VG still did not benefit the intensity of the vortex on both *Γ** and *Γ_LEV_**. The alula-VG produced spanwise flow against the outboard motion of the LEV, which suddenly resulted in an increase in the local strength of the vortex at *r** = 0.71. There was a similar effect at *α* = 25° on the outboard region of the wing (*r** = 0.71 and 0.76). However, at *α* = 45° (the post-stall condition), because of the strong flow interaction mentioned above, the total circulation on the surface was less than that of the base wing model. The results show that the alula-VG had a poor aerodynamic effect at low angles of attack. For the motion at high angles of attack, the better position of the alula-VG was highly dependent on the exact value of the angle of attack. Basically, the model operated at the higher angle of attack should be equipped with an alula-VG inside of the wing model.

### 3.3. The Resultant Sectional Lift Coefficient

By applying the Kutta–Joukowski theorem, the sectional lift coefficients (*C_l_*) at different y-positions were calculated using the total circulation on a chordwise (x-z) plane. The alula-VG enhanced the spanwise flow and influenced the local chordwise circulation, as shown in the previous section. Here, the affected sectional lift coefficient with an alula-VG was considered. Figure 12 shows the differences in the sectional lift coefficients, *ΔC_l_* (= *C_l,VG case_* − *C_l,baseline case_*), between the base wing model and the model with an alula-VG. For the case in which an alula-VG was placed at *L_w_** = 0.25, the change in the *ΔC_l_* was minimal at a low angle of attack (*α* = 10°). For the same wing model at *α* = 25° and 45°, the alula-VG increased the sectional lift at *r** = 0.36–0.51. This is because there was an increase in the strength of spanwise flow and the LEV in the region near the alula-VG.

For the model with an alula-VG placed at the outboard of the wing (*L_w_** = 0.50 or *r** = 0.62), the sectional lift coefficient still showed almost no difference at a small angle of attack (*α* = 10°). At *α* = 25°, there was a decrease in the value of *ΔC_l_* at *r** = 0.66, which was because there was lift-off and bursting of the LEV. For the post-stall condition (*α* = 45°), the value of *ΔC_l_* was negative at *r** = 0.66–0.81, which was also due to the vortex bursting and flow separation on the upper surface of the wing. The variations in the sectional lift coefficients are highly consistent with the previous analysis of the flow field.

## 4. Conclusions

In this study, the aerodynamic effect of an alula-like vortex generator placed on a revolving wing model was investigated. The PIV measurement technique was utilized to visualize and quantitatively analyze the flow structure, which was caused by the motion and geometry of the wing model. The velocity field, the vorticity contour, the circulation of certain vortex structures, and the corresponding sectional lift coefficient were all used to explain the mechanism and the effect of the alula-VG. The results show that, compared to an alula/VG mounted on a fixed-wing model, the flow interactions among the alula-VG induced spanwise flow, the inertial force caused by the revolving motion, and the wing-tip vortex play important roles in the vortex bursting and the resultant aerodynamic performance. Variations in the chordwise circulation and the section lift coefficient correspond to the flux in the spanwise vorticity that depends on the angle of attack for a wing. At a low angle of attack, an attached LEV resulted in a relatively uniform distribution in the chordwise circulation and the sectional lift coefficient for all cases with/without an alula-VG. The effect of the alula-VG depended on its spanwise location and the angle of attack of the wing model. For motion at high angles of attack, the better position of the alula-VG was highly dependent on the angle of attack. Basically, the higher the angle of attack, the further inside a region should the alula-VG be placed.

## Figures and Tables

**Figure 1 biomimetics-07-00128-f001:**
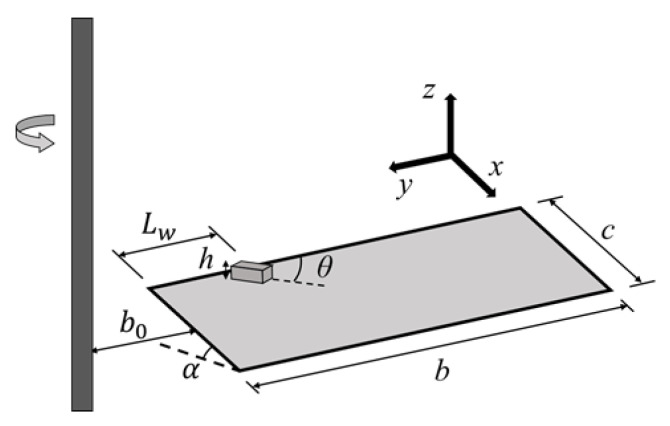
The schematic of the test model with an alula-VG used in this study.

**Figure 2 biomimetics-07-00128-f002:**
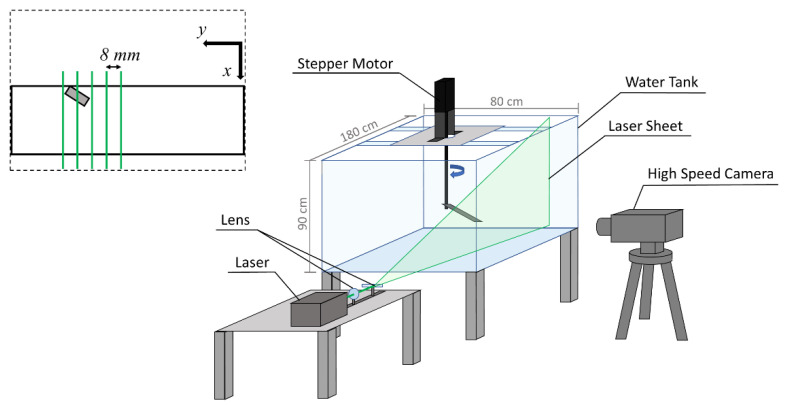
The schematic of the experimental setup. The wing model was mounted on a rotating device and put into a water tank accompanied by a PIV measurement system.

**Figure 3 biomimetics-07-00128-f003:**
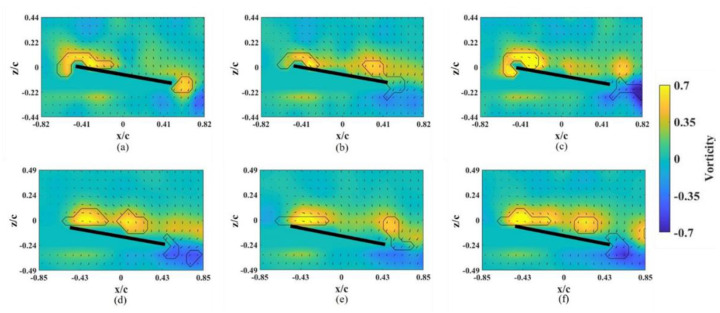
Vorticity contours around the base wing model without alula-VG on different x-z planes at *α* = 10°. (**a**) *r** = 0.36, (**b**) *r** = 0.46, (**c**) *r** = 0.56, (**d**) *r** = 0.66, (**e**) *r** = 0.76, and (**f**) *r** = 0.8.

**Figure 4 biomimetics-07-00128-f004:**
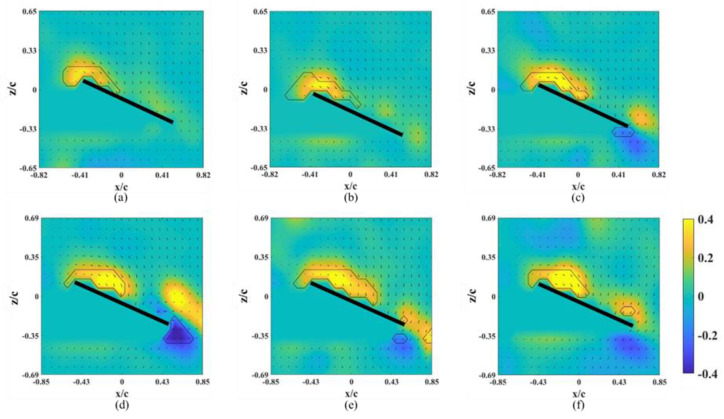
Vorticity contours around the base wing model without alula-VG on different x-z planes at *α* = 25°. (**a**) *r** = 0.36, (**b**) *r** = 0.46, (**c**) *r** = 0.56, (**d**) *r** = 0.66, (**e**) *r** = 0.76, and (**f**) *r** = 0.8.

**Figure 5 biomimetics-07-00128-f005:**
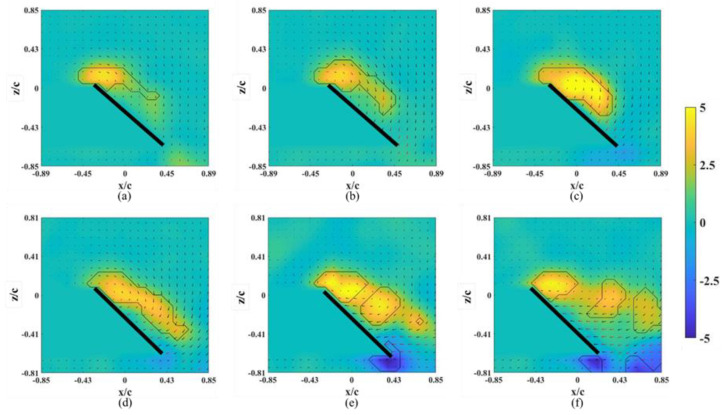
Vorticity contours around the base wing model without alula-VG on different x-z planes at *α* = 45°. (**a**) *r** = 0.36, (**b**) *r** = 0.46, (**c**) *r** = 0.56, (**d**) *r** = 0.66, (**e**) *r** = 0.76, and (**f**) *r** = 0.8.

**Figure 6 biomimetics-07-00128-f006:**
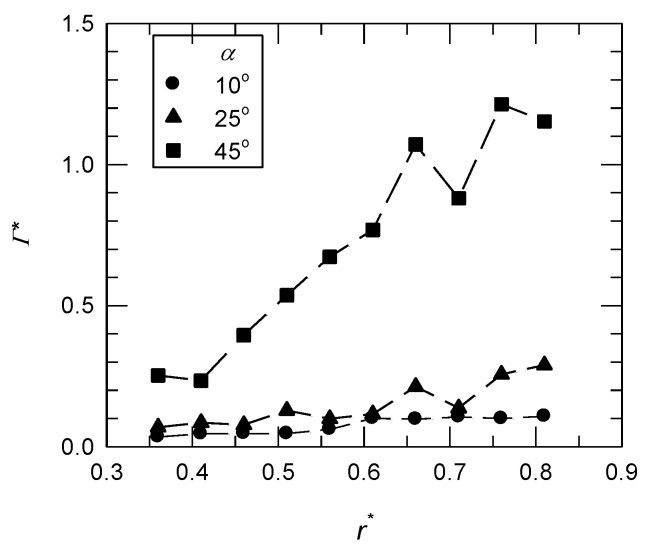
The normalized circulation (*Γ**) of the base wing model on different x-z planes at different angles of attack.

**Figure 7 biomimetics-07-00128-f007:**
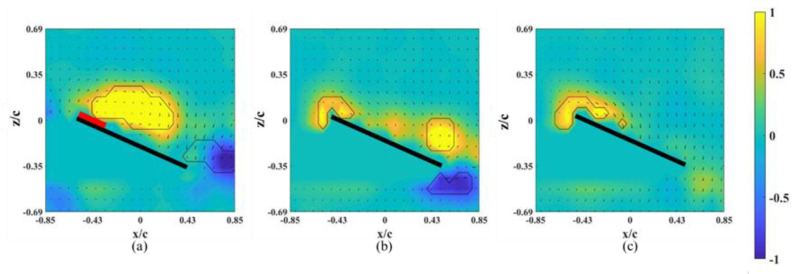
Vorticity contours around the wing model with an alula-VG placed at *L_w_** = 0.25 (*r** = 0.37) and *α =* 25°. (**a**) *r** = 0.36, (**b**) *r** = 0.46, and (**c**) *r** = 0.56.

**Figure 8 biomimetics-07-00128-f008:**
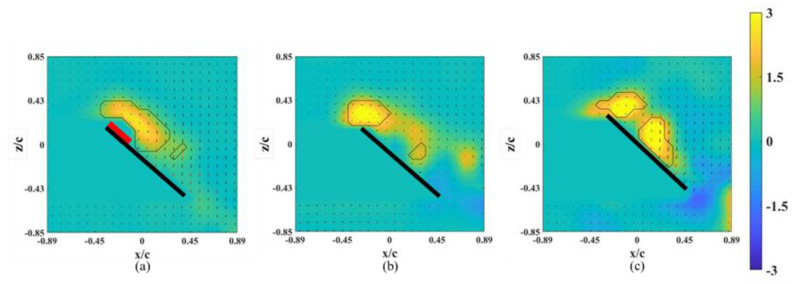
Vorticity contours around the wing model with an alula-VG placed at *L_w_** = 0.25 (*r** = 0.37) and *α =* 45°. (**a**) *r** = 0.36, (**b**) *r** = 0.46, and (**c**) *r** = 0.56.

**Figure 9 biomimetics-07-00128-f009:**
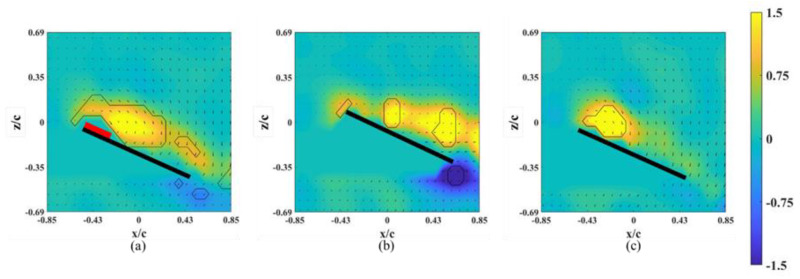
Vorticity contours around the wing model with an alula-VG placed at *L_w_** = 0.50 (*r** = 0.62) and *α =* 25°. (**a**) *r** = 0.61, (**b**) *r** = 0.71, and (**c**) *r** = 0.81.

**Figure 10 biomimetics-07-00128-f010:**
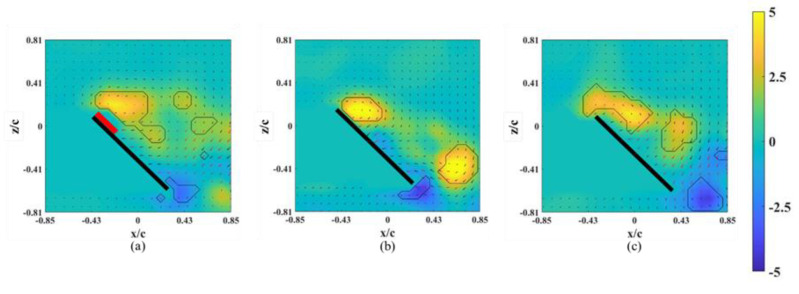
Vorticity contours around the wing model with an alula-VG placed at *L_w_** = 0.50 (*r** = 0.62) and *α =* 45°. (**a**) *r** = 0.61, (**b**) *r** = 0.71, and (**c**) *r** = 0.81.

**Figure 11 biomimetics-07-00128-f011:**
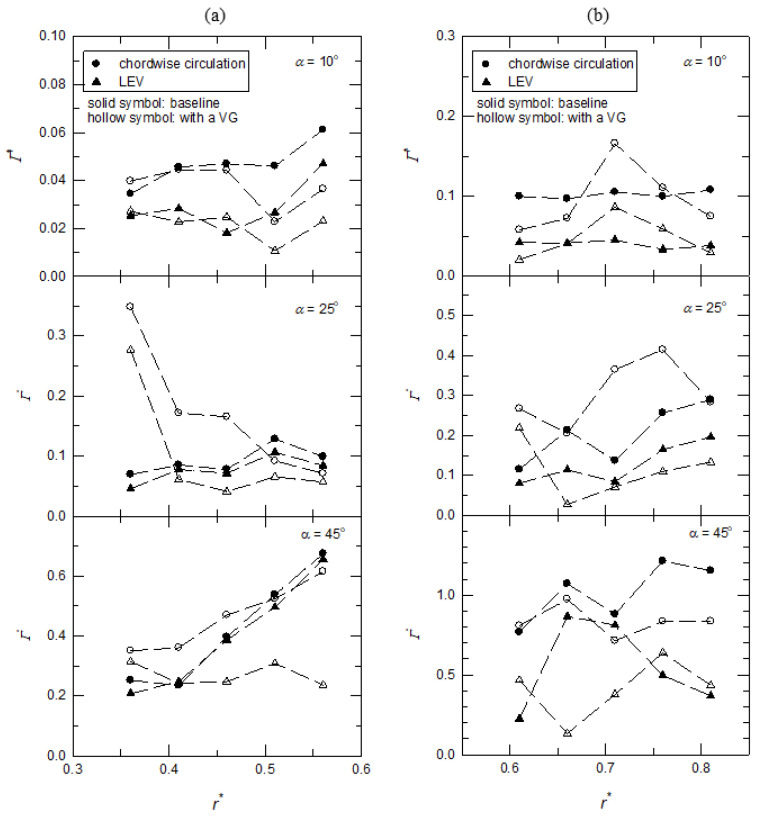
The normalized circulation (*Γ**) of the wing model with an alula-VG on different x-z planes at different angles of attack. (**a**) An alula-VG placed at *L_w_** = 0.25; (**b**) an alula-VG placed at *L_w_** = 0.50.

**Figure 12 biomimetics-07-00128-f012:**
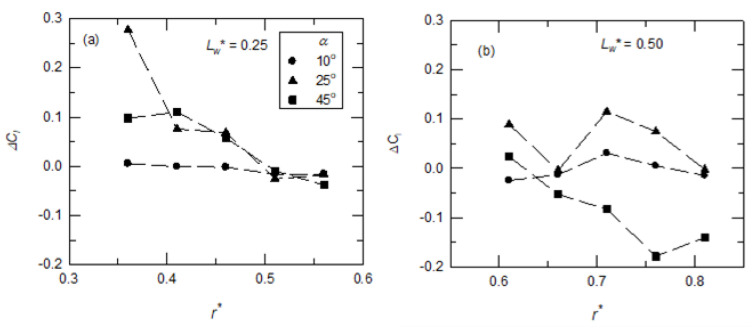
The differences in the sectional lift coefficients (*C_l_*) at different x-z planes at different angles of attack. The wing model was equipped with an alula-VG at different y-positions: (**a**) *L_w_** = 0.25 (*r** = 0.37), (**b**) *L_w_** = 0.50 (*r** = 0.62).

## Data Availability

Not applicable.

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
