# Peer review of "The Aerodynamic Effect of an Alula-like Vortex Generator on a Revolving Wing"

_biomimetics, 2022, doi:10.3390/biomimetics7030128_

Round 1

Reviewer 1 Report

The use of a vortex generator on aerodynamic performance is very interesting. I have read the manuscript of "The Aerodynamic Effect of an Alula-like Vortex Generator on a Revolving Wing." It is good work. However, these comments should be taken into account:

Minor

  1. The introduction needs to be improved. Authors can elaborate more on the introduction of the review, the challenging point of the current model, and highlight this to state their new findings. 
  2. The authors should provide details regarding the geometrical design of the Alula-VG (the Figure can help for better understanding).
  3. There must be more 'explanation' on your motivations for selecting the values of parameters for the model design.
  4. I see the vorticity contours for a = 25° and 45°, but how about a = 10° for the base wing model?
  5. There was no explanation for Figures 3 (a) and (b). The physical mechanism of how the vortex grows when the r* changes should be added.
  6. A similar problem is found in Figures 4, 5, 6, and 7. The discussion must be improved. The authors should give a wider view and describe the reasons for the results. They have to try to show the explanation (or a hypothesis) based on the results they have achieved and presented in the paper.

Author Response

We have carefully checked all the review comments and modified our revised manuscript. Please see the attachment to check the point-by-point response.

Reviewer 2 Report

The paper is well written with minor English grammatical errors.  There were a few places where the reviewer found the writing to be unclear.  All of these are commented on in the attached file.

Author Response

Thank you for the careful correction. We have carefully checked the manuscript and fixed the typos and the grammatical glitches in the revised manuscript. The unclear statement in the Introduction, which was noted by the reviewer, was also rewrote in the revised manuscript.

Round 2

Reviewer 1 Report

Thank you for taking my suggestions. However, I could not find the responses for points 2 and 5.

Author Response

It is our carelessness that we answered Q2&3 and Q5&6 together with unclear descriptions. We have responded to all the questions in detail. Please check the attached file.
